# Electrospun VDF-TeFE Scaffolds Modified by Copper and Titanium in Magnetron Plasma and Their Antibacterial Activity against MRSA

**Arsalan D. Badaraev [1], Marat I. Lerner [2], Dmitrii V. Sidelev [1] , Evgeny N. Bolbasov [1] and Sergei I. Tverdokhlebov [1,*]**

[1] School of Nuclear Science & Engineering, Tomsk Polytechnic University, 30, Lenin Avenue, 634050 Tomsk, Russia; adb6@tpu.ru (A.D.B.); sidelevdv@tpu.ru (D.V.S.); ftoroplast@tpu.ru (E.N.B.)

[2] Institute of Strength Physics and Materials Sciences of Siberian Branch of the Russian Academy of Sciences, 2/4, Akademicheskii Avenue, 634055 Tomsk, Russia; lerner@ispms.tsc.ru

* Correspondence: tverd@tpu.ru

**Abstract:** Copolymer solution of vinylidene fluoride with tetrafluoroethylene (VDF-TeFE) was used for electrospinning of fluoropolymer scaffolds. Magnetron co-sputtering of titanium and copper targets in the argon atmosphere was used for VDF-TeFE scaffolds modification. Scanning electron microscopy (SEM) showed that scaffolds have a nonwoven structure with mean fiber diameter $0.77 \pm 0.40$ μm, mean porosity $58 \pm 7\%$. The wetting angle of the original (unmodified) hydrophobic fluoropolymer scaffold after modification by titanium begins to possess hydrophilic properties. VDF-TeFE scaffold modification by titanium/copper leads to the appearance of strong antibacterial properties. The obtained fluoropolymer samples can be successfully used in tissue engineering.

**Keywords:** electrospinning; scaffold; fluoropolymer; plasma modification; titanium; copper; methicillin-resistant St. aureus; antibacterial activity





## 1. Introduction

Copolymer of vinylidene fluoride with tetrafluoroethylene (VDF-TeFE) has high chemical and thermal resistance, ability to dissolve in organic solvents [1] and piezoelectric properties [2]. The use of materials with piezoelectric properties leads to stimulation of bone growth and differentiation of nerve cells [3]. Due to this, biocompatible piezoelectric fluoropolymers are actively used in the manufacture of tissue-engineered scaffolds [4].

The electrospinning method is actively used in polymeric scaffold production for tissue engineering needs [5]. Electrospinning materials are characterized by relatively high mechanical properties and biocompatibility. This is due to the possibility of manufacturing fibers from 10 nm to 10 μm [6], with extremely high surface area [5] and the ability to imitate the topology of extracellular matrix [7].

Despite this, Electrospun VDF-TeFE scaffolds are hydrophobic and unable to negatively affect the number of pathogenic bacteria, which limits their use in tissue engineering. Application of drug loaded scaffolds can stimulate the growth of antibiotic-resistant pathogenic bacteria. The advantage of using metals as an antimicrobial agent is their ability to negatively affect the number of bacteria with antibiotic resistance [8].

It has previously been shown that the magnetron sputtering of copper allows to preserve the original morphology and surface structure of the nonwoven fluoropolymers and makes it possible to impart antimicrobial properties [9,10]. Compared to chemical vapor deposition and plasma electrolytic oxidation, modification by magnetron sputtering can produce high-quality metal containing films [11–13].

Nevertheless, copper has toxic [14] and hydrophobic properties [15]. To increase the wettability and biocompatibility of polymeric scaffolds, plasma modification by titanium is used [16].

Magnetron co-sputtering can be used to create composite coatings which consist of several metallic components. Thin films of copper and titanium formed by magnetron co-sputtering have hydrophilic and strong antibacterial properties [17]. Scaffold modification by mixed flows of copper and titanium could potentially possess antibacterial and high wetting properties.

The aim of this work is the manufacture of VDF-TeFE scaffolds, modified by copper and titanium, which will have antibacterial properties against methicillin-resistant St. aureus (MRSA) for tissue engineering needs.

## 2. Materials and Methods

### 2.1. Fabrication of VDF-TeFE Scaffolds

Fluoropolymer scaffolds were formed from 5% solution of VDF-TeFE (Galopolymer, Moscow, Russia) in acetone ($C_3H_6O$, Acrus, Moscow, Russia) by the electrospinning method on installation NANON-01A (MECC Co., Fukuoka, Japan). The folowing technological modes of installation were used: voltage and distance between collector and needle 20 kV and 150 mm, respectively; cylindrical collector 100 mm in diameter and 200 mm in length; collector rotation speed 200 r/min, and flow rate of polymeric solution 6 mL/h.

To increase the tensile strength, VDF-TeFE scaffolds were placed in a drying oven at normal atmospheric pressure at 130 °C for 12 h.

### 2.2. Plasma Modification of VDF-TeFE Scaffolds

VDF-TeFE scaffolds were modified in the installation (Figure 1) [18] by using magnetron co-sputtering of copper (Cu, 99.95%) and titanium (Ti, 99.95%) targets. During the modification, the following materials and conditions were used: circular Ti and Cu targets with 90 mm in diameter and 8 mm in height, argon as working gas (Ar, 99.998%), operation pressure 0.3 Pa.

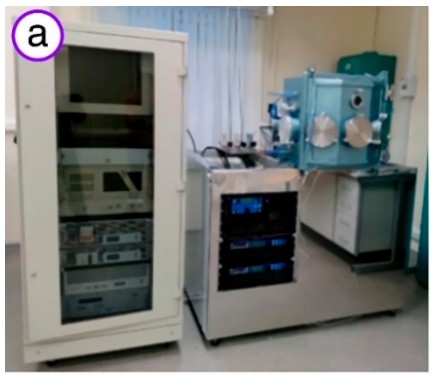 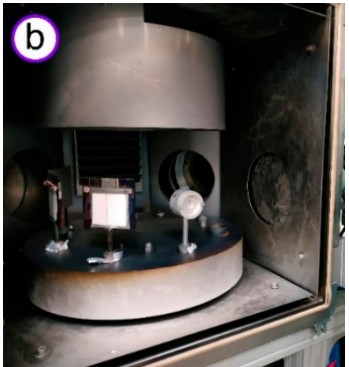

**Figure 1.** Photos of magnetron installation (**a**) and its vacuum chamber with loaded samples (**b**).

The technological modes of VDF-TeFE scaffolds plasma modification are shown in Table 1.

**Table 1.** Magnetron modification modes of vinylidene fluoride with tetrafluoroethylene (VDF-TeFE) scaffolds by titanium and copper.

| Modification Modes (Samples) | Discharge Power, W | | Current, A | | Modification Time, min |
|---|---|---|---|---|---|
| | Cu | Ti | Cu | Ti | |
| 100 Ti | - | 750 | - | 1.5 | 32.6 |
| Ti/Cu 1 | 65 | 750 | 0.15 | 1.5 | 26.6 |
| Ti/Cu 2 | 130 | 500 | 0.30 | 1.1 | 29.3 |

Co-sputtering of copper and titanium targets were provided by the unipolar power supply units APEL-M-5PDC (JCS "Applied Electronics", Tomsk, Russia) with 100 kHz frequency and 70% duty cycle.

Modes of plasma modification were chosen for deposition of metallic films with the same thickness onto metallic plates (used as the reference substrate). The thickness of thin metallic films was evaluated on a quartz thickness gauge Mikron-5 (Izovak, Minsk, Belarus). It was shown that the mean thickness of all deposited thin films was ~200 ± 30 nm.

### 2.3. Characterization of VDF-TeFE Scaffolds

#### 2.3.1. Scanning Electron Microscopy

The morphology of VDF-TeFE scaffolds was investigated on a JCM-6000 Plus instrument (Jeol, Akishima, Japan) by method scanning electron microscopy (SEM). Scaffold surface images were taken at 1000× magnification. To increase the electrical conductivity of samples, a thin golden layer was deposited on the VDF-TeFE scaffold surfaces using a SmartCoater device (Jeol, Akishima, Japan). Surface porosity and fiber diameters were estimated using the ImageJ 1.48 program (National Institute of Health, Washington, DC, USA) with plug-in DiameterJ v1.018 (National Institute of Standards and Technology, Gaithersburg, MD, USA).

#### 2.3.2. Energy Dispersive X-ray Spectroscopy

Elemental composition of VDF-TeFE scaffolds was studied by energy dispersive X-ray spectroscopy (EDAX) on a JCM-6000 Plus instrument (Jeol, Akishima, Japan). The collected data of the elements' concentration was corrected using ZAF corrections. Accelerating voltage of the beam was 15 kV, beam current 1 nA, and sample analysis mean time ~130 s.

#### 2.3.3. Wettability

The contact angle values were measured by sessile drop method on a DSA-25 set up (KRÜSS, Hamburg, Germany). Droplets of deionized water (2 μL) were placed in different locations on the scaffold and images were captured after 1 min disposition of each drop.

#### 2.3.4. Mechanical Properties

Mechanical properties of the VDF-TeFE scaffolds were evaluated on an Instron 3343 instrument (Illinois Tool Works, Glenview, IL, USA) with an Instron 2519-102 sensor (Illinois Tool Works, Glenview, IL, USA). The traverse speed was set at 10 mm/min, the sample size was 30 × 10 mm, and the length of the sample testing area was 10 mm.

#### 2.3.5. Antibacterial Activity

For the testing of antibacterial ability of obtained scaffolds, the standard JIS L 1902/ISO 20,743 "Determination of Antibacterial Activity of Antibacterial Finished Products (Textiles)" was used [10]. Antimicrobial activity was investigated against methicillin-resistant St. aureus (MRSA) strain AATCC 43300.

The calculation of the antibacterial activity indicator (R) was carried out according to the formula:

$$R = 100 \times (C - A)/C \tag{1}$$

where A is the number of bacteria isolated from inoculated modified by plasma samples incubated over time; and C is the number of bacteria isolated from the control samples immediately after inoculation.

### 2.4. Statistical Analysis

Statistical data processing was performed using the OriginPro® 2019 program (OriginLab, Northampton, MA, USA). Differences in fiber diameters and wettability were evaluated using the Mann-Whitney U test. The differences were statistically significant at $p < 0.05$.

## 3. Results

### 3.1. Scanning Electron Microscopy and Energy Dispersive X-ray Spectroscopy

SEM images at $1000\times$ magnification, histograms of fiber diameter and EDAX spectra of VDF-TeFE scaffolds modified by magnetron plasma by copper and titanium are presented in Figure 2.

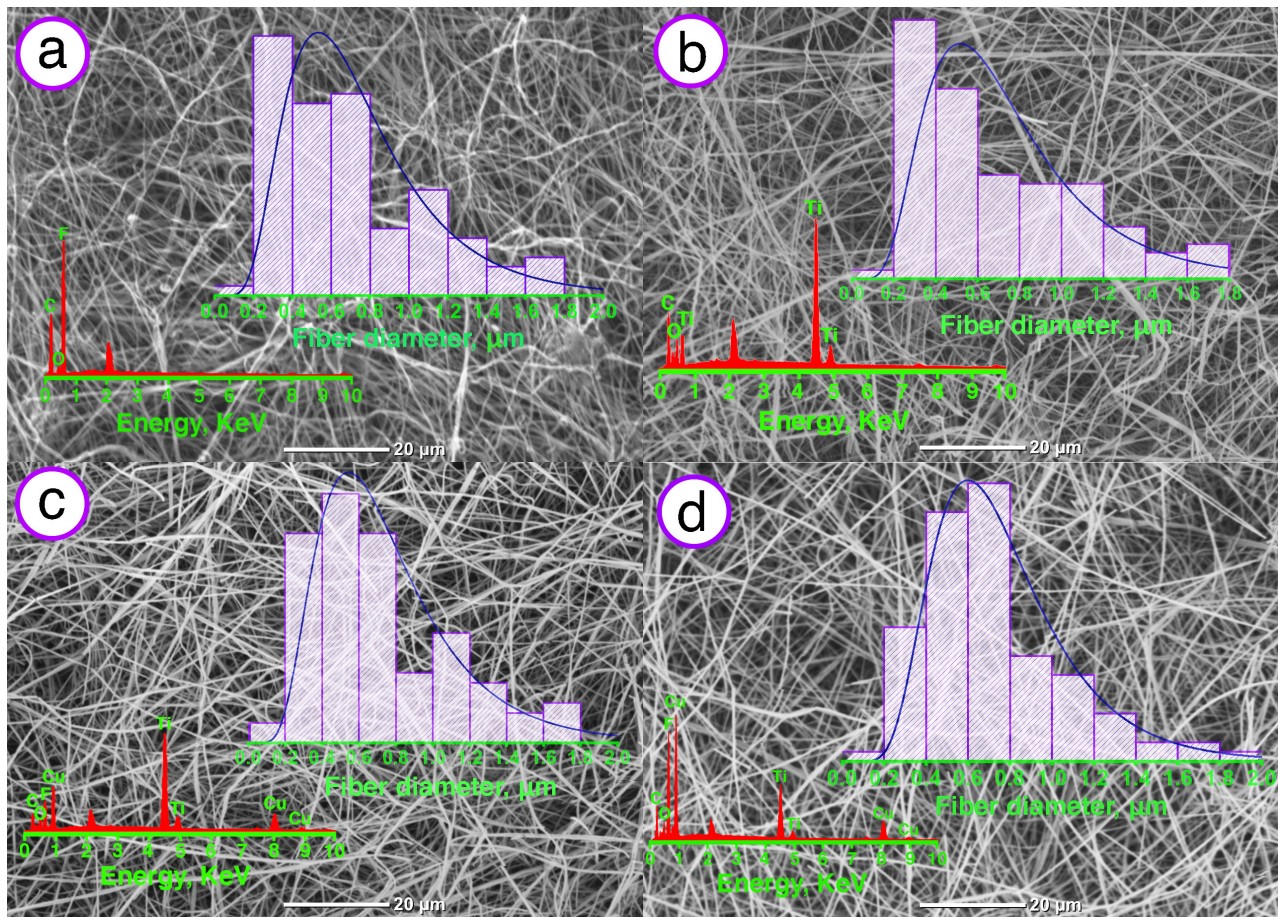

**Figure 2.** SEM images, fiber diameter histograms and energy dispersive X-ray spectroscopy (EDAX) spectra of: (**a**) Pristine (unmodified VDF-TeFE scaffold); (**b**) Ti; (**c**) Ti/Cu 1; (**d**) Ti/Cu 2.

VDF-TeFE scaffolds have a characteristic nonwoven structure. Scaffolds consist of numerous chaotically interlaced polymer threads. Plasma modification of scaffolds in various modes allows the preservation of the pristine surface morphology.

EDAX spectra have shown that pristine scaffolds consist of carbon (C), fluorine (F) and oxygen (O). The peak of gold at 2.1 keV is also observed. After plasma modification, the titanium (Ti) and copper (Cu) peaks with high intensity are revealed.

Mean surface porosity and fiber diameter of VDF-TeFE scaffolds modified by magnetron co-sputtering of titanium and copper target are demonstrated in Table 2.

**Table 2.** Surface porosity and fiber diameter of plasma modified VDF-TeFE scaffolds.

| Sample | Surface Porosity, % | Fiber Diameter, μm |
|---|---|---|
| Pristine | $58 \pm 7$ | $0.76 \pm 0.37$ |
| 100 Ti | $61 \pm 6$ | $0.75 \pm 0.38$ |
| Ti/Cu 1 | $62 \pm 6$ | $0.79 \pm 0.37$ |
| Ti/Cu 2 | $55 \pm 9$ | $0.78 \pm 0.32$ |

The pristine scaffold has porosity 58 ± 7% and mean fiber diameter 0.76 ± 0.40 μm. Magnetron plasma modification of fluoropolymer scaffolds by titanium and titanium/copper allows the maintenance of the porosity values and fiber diameters of the pristine scaffold.

Values of the elements atomic concentration and their ratio in VDF-TeFE scaffolds modified by titanium and copper are presented in Table 3.

**Table 3.** Elements atomic concentration and ratio of plasma modified VDF-TeFE scaffolds.

| Sample | Elemental Concentration, atom. % | | | | | Elemental Ratio | |
|---|---|---|---|---|---|---|---|
| | C | O | F | Ti | Cu | C/O | Ti/Cu |
| Pristine | 55.1 | 5.6 | 39.3 | - | - | 10.4 | - |
| Ti | 36.8 | 20.7 | 34.3 | 8.2 | - | 1.8 | - |
| Ti/Cu 1 | 37.4 | 15.8 | 29.2 | 10.3 | 7.3 | 2.4 | 1.4 |
| Ti/Cu 2 | 43.1 | 12.0 | 31.0 | 5.8 | 8.1 | 3.6 | 0.7 |

After plasma modification of VDF-TeFE, the content of carbon and fluorine decreases, the content of oxygen increases and the elements titanium and copper are observed.

The pristine sample has extremely high C/O ratio. Modification of fluoropolymer scaffolds by titanium strongly decrease (by 5.8 times) the C/O ratio. Modification in modes Ti/Cu 1 and Ti/Cu 2 decrease the C/O ratio by 4.3 and 2.9 times, respectively.

### 3.2. Wettability

The results of the contact angle measurements of VDF-TeFE scaffolds modified by plasma magnetron discharge are presented in Figure 3.

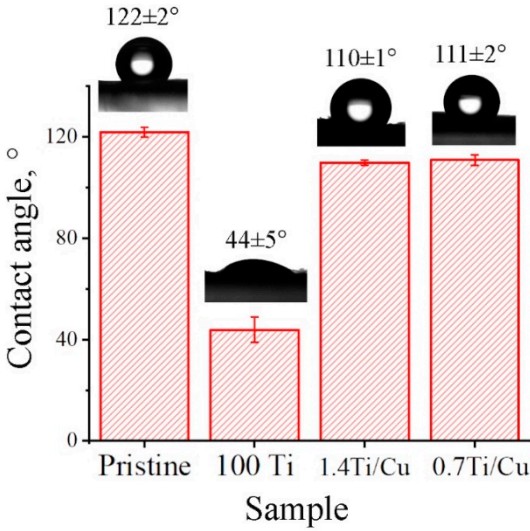

**Figure 3.** Water contact angle values of modified VDF-TeFE sample surfaces.

The wetting angle of the pristine sample has a value of 122 ± 2°, and after plasma modification by titanium, the contact angle of the VDF-TeFE scaffold decreases to 44 ± 5°, which indicates the imparting of hydrophilic properties. Modification by modes Ti/Cu 1 and Ti/Cu 2 leads to insignificant decrease in the water contact angle to 109–113°.

### 3.3. Mechanical Properties

Stress-strain curves and mechanical characteristic values of plasma modified VDF-TeFE scaffolds by copper and titanium are shown in Figure 4 and Table 4, respectively.

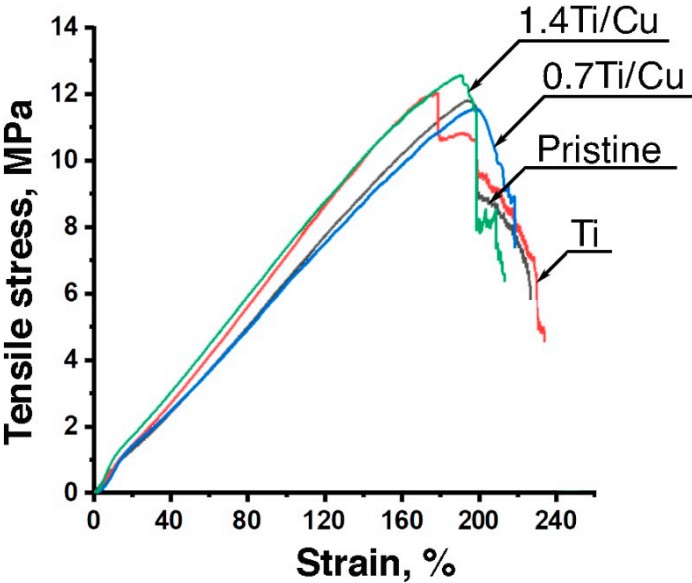

**Figure 4.** Stress-strain curves of plasma modified VDF-TeFE scaffolds.

**Table 4.** Mechanical properties of plasma modified VDF-TeFE scaffolds.

| Sample | Tensile Strength, MPa | Relative Elongation, % | Young's Modulus, MPa |
|---|---|---|---|
| Pristine | 11.8 ± 0.8 | 226 ± 16 | 13.6 ± 0.9 |
| Ti | 12.0 ± 0.4 | 230 ± 18 | 15.3 ± 2.1 |
| Ti/Cu 1 | 11.6 ± 0.5 | 218 ± 24 | 13.2 ± 0.3 |
| Ti/Cu 2 | 12.6 ± 0.7 | 214 ± 22 | 15.1 ± 1.1 |

The stress-strain curves are almost straight until the maximum tensile stresses are reached.

The pristine sample has a tensile strength of 11.8 ± 0.8 MPa, relative elongation 226 ± 16% and Young's modulus 13.6 ± 0.9 MPa. Plasma modification by titanium and titanium/copper allowed the preservation of the mechanical values of the pristine VDF-TeFE scaffold.

### 3.4. Antibacterial Activity

Photographs of Petri dishes containing MRSA, number of bacteria and antibacterial activity indicator (R) of VDF-TeFE scaffolds are presented in Figure 5 and Table 5.

**Table 5.** Results of antibacterial activity assessment.

| Sample | Number of Bacteria, CFU/mL | R, % |
|---|---|---|
| Pristine | $(6 \pm 0.5) \times 10^6$ | - |
| Ti | $(5.2 \pm 0.9) \times 10^6$ | 13 |
| Ti/Cu 1 | $(5.6 \pm 0.6) \times 10^5$ | 91 |
| Ti/Cu 2 | $(2.7 \pm 0.8) \times 10^5$ | 96 |

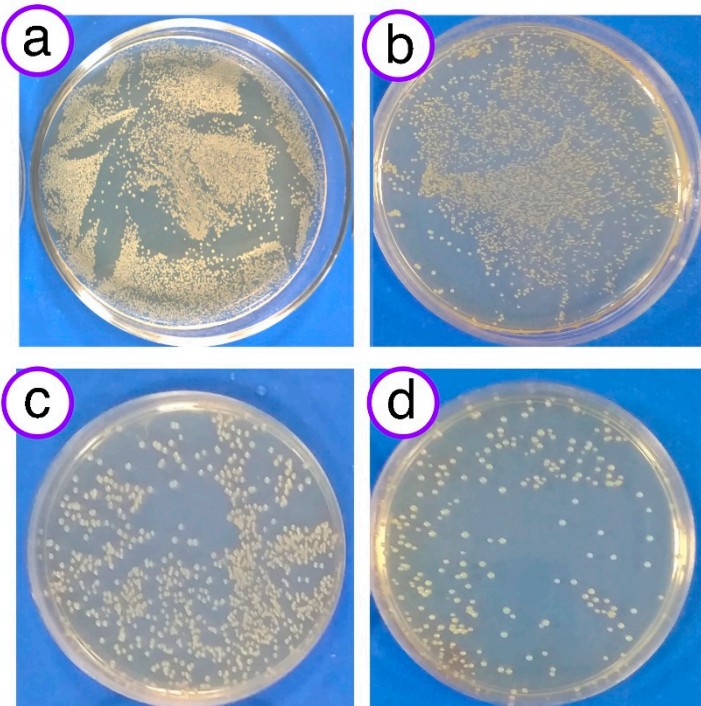

**Figure 5.** Photographs of Petri dishes containing methicillin-resistant St. aureus (MRSA) colony forming units (CFUs) on agar: (**a**) Pristine (unmodified VDF-TeFE scaffold); (**b**) Ti; (**c**) Ti/Cu 1; (**d**) Ti/Cu 2.

Pristine and Ti modified scaffolds do not negatively affect the number of MRSA, indicating the absence of antibacterial properties. In the samples modified with titanium/copper, the number of bacteria was less than 90% compared to the original sample. This indicates the strong antibacterial activity of VDF-TeFE scaffolds modified by Ti/Cu.

## 4. Discussion

### 4.1. Influence of Plasma Modification by Ti/Cu on Morphology, Wettability and Mechanical Properties of VDF-TeFE Scaffolds

We have shown earlier that plasma modification by magnetron co-sputtering of nonwoven materials from VDF-TeFE allows the maintenance of the original fiber diameter and mechanical properties [9]. This is the reason for the preservation of the morphology and surface structure, as in the pristine sample.

It was noted that the modification of synthetic polymer scaffolds with titanium by magnetron co-sputtering makes it possible to increase the surface wettability and impart hydrophilic properties [16]. A decrease in contact angles for VDF-TeFE scaffolds modified by Ti/Cu in comparison with the pristine sample by ~10° are associated with the presence of titanium on the surface and with the hydrophobic nature of copper.

An almost straight stress-strain curve shape was found in [19], where mechanical properties of a fluoropolymer membrane manufactured by the electrospinning method were researched. The retention of mechanical properties may be due to the fact that during the plasma treatment of scaffolds only subsurface fibers are modified, while the fibers located deep in the volume of the scaffold remain intact [9].

### 4.2. Influence of Plasma Modification by Ti/Cu on Elemental Composition of VDF-TeFE Scaffolds

A significant decrease in the C/O ratio after plasma modification is primarily associated with a significant increase in the oxygen content. The surface of VDF-TeFE scaffolds becomes chemically active after modification. Therefore, when scaffolds are removed from the vacuum chamber of the plasma co-sputtering unit, their surface begins to absorb oxygen

from the air which leads to the formation of oxygen-containing chemical compounds [20]. Compounds of copper, titanium and carbon with oxygen appear most likely [9].

The smaller sample C/O ratio modified only with titanium in comparison with the Ti/Cu 1 and Ti/Cu 2 samples is due to the high oxygen content. With a decrease in the Ti/Cu ratio in modified scaffolds, C/O also increases. These factors indicate a more active absorption of oxygen by the titanium-modified surface than by modified Ti/Cu. Perhaps this is because titanium reacts more actively with oxygen than copper.

The presence of oxygen in the pristine sample is associated with placing the scaffold in a drying oven without evacuating the atmosphere, facilitating the interaction of oxygen from the air with carbon from the polymer.

### 4.3. Influence of Plasma Modification by Ti/Cu on Antibacterial Properties of VDF-TeFE Scaffolds

It is known that plasma modification of VDF-TeFE scaffolds with copper makes antibacterial properties possible. Modification of scaffolds with mixed flows of titanium and copper does not prevent antibacterial properties in copper. The higher index of antibacterial activity in the Ti/Cu 1 sample as compared to Ti/Cu 2 is associated with the higher concentration of copper on the surface.

### 5. Conclusions

Electrospun VDF-TeFE scaffolds were modified by magnetron plasma discharge by titanium and titanium/copper. Plasma modification does not affect the morphology and mechanical properties of VDF-TeFE scaffolds but changes the elemental composition. Modification of fluoropolymer scaffolds by titanium makes it possible to impart hydrophilic properties. The contact angle for the Ti sample is $44 \pm 5°$. Pristine scaffold modification by mixed Ti/Cu flows reduces the water contact angle by ~10°. Modification by Ti/Cu allows scaffolds to have strong antibacterial properties. The amount of MRSA on Ti/Cu scaffolds is 90% less than on pristine scaffolds.

Due to the presence of antibacterial properties (Ti/Cu 1, Ti/Cu 2) and hydrophilicity (Ti sample), obtained scaffolds can be successfully applied in tissue engineering. The creation of a scaffold with hydrophilic and antibacterial properties will be the subject of further research.

**Author Contributions:** Conceptualization. A.D.B.; methodology. E.N.B., S.I.T.; investigation. A.D.B., M.I.L., D.V.S.; writing—original draft preparation. A.D.B.; writing—review and editing. S.I.T.; supervision. S.I.T.; funding acquisition. S.I.T. All authors have read and agreed to the published version of the manuscript.

**Funding:** Financially supported by Tomsk Polytechnic University Competitiveness Enhancement Program project VIU-SEC B.P. Veinberg-196/2020 (S.I.T.).

**Institutional Review Board Statement:** Not applicable.

**Informed Consent Statement:** Not applicable.

**Acknowledgments:** The authors are grateful to the Resource Centre "Materials Science Shared Center" part of the "Tomsk Regional Common Use Center (TRCUC)" of Tomsk State University for providing measurements and Olga Bakina for conducting antibacterial activity study, which was performed according to the Government research assignment for ISPMS SB RAS, project FWRW-2021-0007.

**Conflicts of Interest:** The authors declare no conflict of interest.

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
