# Peer review of "Electrospun VDF-TeFE Scaffolds Modified by Copper and Titanium in Magnetron Plasma and Their Antibacterial Activity against MRSA"

_technologies, doi:10.3390/technologies9010005_

Round 1
Reviewer 1 Report
Comments:
In this study, Electrospun VDF-TeFE scaffolds modified by copper and titanium in magnetron plasma and their antibacterial activity against MRSA. This paper is probably publishable however; the author should address the following questions.
1) Please modify the coloring of the labelings in SEM and other graph to enhance the visibility.
2) The author should separate the graphs of fiber diameter histograms and contact angle measurement and label them as Fig.2 and Fig.3
3) How the author calculated the number of pores using ImageJ, as there’s no specific pores can be seen on the surface
- The author should include the EDS graphical results
- The author should draw the mechanical result in graphic form.
- Please include the article in the coating introductions,
- Zeeshan Ur Rehman, Bon Heun Koo ,Yeon-Gil Jung ,Je Hyun Lee and Dongjin Choi, Effect of K2ZrF6 Concentration on the Two-Step PEO Coating Prepared on AZ91 Mg Alloy in Alkaline Silicate Solution, Materials 2020, 13(3), 499; https://doi.org/10.3390/ma13030499.
- Zeeshan Ur Rehman,Bon Heun Koo, and Dongjin Choi Influence of Complex SiF62− Ions on the PEO Coatings Formed on Mg–Al6–Zn1 Alloy for Enhanced Wear and Corrosion Protection, Coatings 2020, 10(2), 94; https://doi.org/10.3390/coatings10020094.
- Anwarul Hasan, Adnan Memic, Nasim Annabi Monowar Hossain Arghya Paula Mehmet R. Dokmeci Fariba Dehghani Ali Khademhosseini, Electrospun scaffolds for tissue engineering of vascular grafts, Acta Biomaterialia, 10, 2014, 11-25
Author Response
Responses to Reviewer’s comments
Reviewer #1:
In this study, Electrospun VDF-TeFE scaffolds modified by copper and titanium in magnetron plasma and their antibacterial activity against MRSA. This paper is probably publishable however; the author should address the following questions.
Comment #1: Please modify the coloring of the labelings in SEM and other graph to enhance the visibility.
Answer: We thank the reviewer for this comment. The coloring of the labelings in SEM and other graph were modified.
Comment #2: The author should separate the graphs of fiber diameter histograms and contact angle measurement and label them as Fig.2 and Fig.3
Answer: We thank the reviewer for this comment. The graphs of fiber diameter histograms and contact angle measurement are separated.
Comment #3: How the author calculated the number of pores using ImageJ, as there’s no specific pores can be seen on the surface?
Answer: The estimated pore count has been removed as it does not provide any useful information. To calculate the number of pores a black-white SEM image with maximum contrast was used (figure 1). The pores are clearly visible in this image.
Figure 1. SEM image of pristine sample processed by ImageJ program and diameterJ plug-in
Comment #4: The author should draw the mechanical result in graphic form.
Answer: We thank the reviewer for this comment. The mechanical result was drawn in graphic form.
Comment #5: Please include the article in the coating introductions:
- Zeeshan Ur Rehman, Bon Heun Koo ,Yeon-Gil Jung ,Je Hyun Lee and Dongjin Choi, Effect of K2ZrF6 Concentration on the Two-Step PEO Coating Prepared on AZ91 Mg Alloy in Alkaline Silicate Solution, Materials 2020, 13(3), 499; https://doi.org/10.3390/ma13030499.
- Zeeshan Ur Rehman,Bon Heun Koo, and Dongjin Choi Influence of Complex SiF62− Ions on the PEO Coatings Formed on Mg–Al6–Zn1 Alloy for Enhanced Wear and Corrosion Protection, Coatings 2020, 10(2), 94; https://doi.org/10.3390/coatings10020094.
- Anwarul Hasan, Adnan Memic, Nasim Annabi Monowar Hossain Arghya Paula Mehmet R. Dokmeci Fariba Dehghani Ali Khademhosseini, Electrospun scaffolds for tissue engineering of vascular grafts, Acta Biomaterialia, 10, 2014, 11-25
Answer: The above articles were added in the introduction.

Reviewer 2 Report
The main purpose of the electrospun scaffolds modification was to provide antibacterial efficiency. The objective is not achieved, because one order improvement is far not enough for successful biomedical applications. The number of bacteria still present on the surface is too high and can results in biofilm formation.
In further, no sufficient details are provided on the antibacterial efficiency of the scaffolds, no graphical prove is given, only table with the results which is not sufficient.
In addition, there are very limited procedures on the scaffolds processing and the authors have not revealed and optimized the treatment parameters.
I would not recommend the manuscript to be published in the journal before additional series of experiments are done.
Author Response
Responses to Reviewer’s comments
Reviewer #2:
Comment #1: The main purpose of the electrospun scaffolds modification was to provide antibacterial efficiency. The objective is not achieved, because one order improvement is far not enough for successful biomedical applications. The number of bacteria still present on the surface is too high and can results in biofilm formation.
Answer: We agree that the antibacterial activity of obtained VDF-TeFE scaffolds modified by Ti and Cu is not high enough, as we would like for future applications. Nevertheless, the main goal of the article was achieved, since in the work we show the possibility of forming a copper-modified scaffolds with antibacterial properties. It would be wrong to say that the samples do not have antibacterial properties, since the number of bacteria on Ti/Cu samples decreases by more than 90%, which is significant. The creation of antibacterial effective and hydrophilic polymeric scaffolds will be the subject of further research.
Comment #2: In further, no sufficient details are provided on the antibacterial efficiency of the scaffolds, no graphical prove is given, only table with the results which is not sufficient
Answer: We thank the reviewer for this comment. The results of antibacterial properties in graphical form were added.
Comment #3: In addition, there are very limited procedures on the scaffolds processing and the authors have not revealed and optimized the treatment parameters.
Answer: The type of our manuscript is "Communication” therefore, we did not provide some of the procedures that were described in other papers and which we plan to publish in the new article. Earlier [Badaraev, A. D., et al. "Magnetron plasma modification by sputtering copper target of electrospun fluoropolymer material to possess bacteriostatic properties." Materials Today: Proceedings 22 (2020): 219-227., Badaraev, A. D., et al. "Piezoelectric polymer membranes with thin antibacterial coating for the regeneration of oral mucosa." Applied Surface Science 504 (2020): 144068.], we have already modified scaffolds from vinylidene fluoride by method magnetron sputtering of copper target in argon atmosphere. Our laboratory has long been engaged in the manufacture of medical materials from VDF-TeFE polymer [Tverdokhlebov, Sergey Ivanovich, et al. "Research of the surface properties of the thermoplastic copolymer of vinilidene fluoride and tetrafluoroethylene modified with radio-frequency magnetron sputtering for medical application." Applied surface science 263 (2012): 187-194.]. We carried out the optimization of modification parameters, the results of which will be published in another article, which is at the printing stage. A part of this article that describes the optimization of the mode Ti/Cu 2 is attached.
References
- Badaraev, A. D., et al. "Magnetron plasma modification by sputtering copper target of electrospun fluoropolymer material to possess bacteriostatic properties." Materials Today: Proceedings22 (2020): 219-227.
- Badaraev, A. D., et al. "Piezoelectric polymer membranes with thin antibacterial coating for the regeneration of oral mucosa." Applied Surface Science504 (2020): 144068.
- Tverdokhlebov, Sergey Ivanovich, et al. "Research of the surface properties of the thermoplastic copolymer of vinilidene fluoride and tetrafluoroethylene modified with radio-frequency magnetron sputtering for medical application." Applied surface science263 (2012): 187-194.
- Bolbasov, E. N., et al. "The investigation of the production method influence on the structure and properties of the ferroelectric nonwoven materials based on vinylidene fluoride–tetrafluoroethylene copolymer." Materials Chemistry and Physics182 (2016): 338-346.

Round 2
Reviewer 1 Report
The manuscript can be accepted in the current form.